# Host Identity and Geographic Location Significantly Affect Gastrointestinal Microbial Richness and Diversity in Western Lowland Gorillas (*Gorilla gorilla gorilla*) under Human Care

**DOI:** 10.3390/ani11123399

**Published:** 2021-11-28

**Authors:** Katrina Eschweiler, Jonathan B. Clayton, Anneke Moresco, Erin A. McKenney, Larry J. Minter, Mallory J. Suhr Van Haute, William Gasper, Shivdeep Singh Hayer, Lifeng Zhu, Kathryn Cooper, Kimberly Ange-van Heugten

**Affiliations:** 1Department of Nutrition, Denver Zoo, Denver, CO 80205, USA; KEschweiler@denverzoo.org; 2Department of Animal Science, NC State University, Raleigh, NC 27695, USA; 3Department of Biology, University of Nebraska at Omaha, Omaha, NE 68182, USA; jclayton@unomaha.edu (J.B.C.); shayer@unomaha.edu (S.S.H.); 4Department of Food Science and Technology, University of Nebraska-Lincoln, Lincoln, NE 68588, USA; mvanhaute@unl.edu; 5Nebraska Food for Health Center, University of Nebraska-Lincoln, Lincoln, NE 68588, USA; 6Primate Microbiome Project, University of Nebraska-Lincoln, Lincoln, NE 68588, USA; 7Department of Animal Welfare and Research, Denver Zoo, Denver, CO 80205, USA; moresco2@gmail.com; 8Department of Clinical Sciences, College of Veterinary Medicine, NC State University, Raleigh, NC 27607, USA; jb.minter@nczoo.org; 9Department of Applied Ecology, NC State University, Raleigh, NC 27695, USA; eamckenn@ncsu.edu; 10Hanes Veterinary Medical Center, North Carolina Zoo, Asheboro, NC 27205, USA; 11College of Information Science and Technology, University of Nebraska at Omaha, Omaha, NE 68182, USA; kgasper@unomaha.edu; 12College of Life Sciences, Nanjing Normal University, Nanjing 210023, China; lzhu@unomaha.edu (L.Z.); kmcooper@unomaha.edu (K.C.)

**Keywords:** gastrointestinal (GIT) microbiome, feces, human managed populations, western lowland gorilla

## Abstract

**Simple Summary:**

Since the advent of microbiome research, this field has seen an explosion of both techniques and subfields. Researchers have aimed not only to classify microbiome membership and diversity among varying hosts, but to also identify and understand new and novel microbial lineages. This wealth of knowledge continues to grow, and with it the potential to use microbiome databases as diagnostic tools. This diagnostic application is of great importance and interest in zoological settings, as it may provide a non-invasive assessment of animal health. However, before this tool can be utilized in zoos, more data are needed to assess the extent of microbial variation characteristics to each host species to know what may be problematic versus normal. The aim of this research was to characterize variation of the microbiome at the individual level within managed populations of western lowland gorillas in three zoological institutions.

**Abstract:**

The last few decades have seen an outpouring of gastrointestinal (GI) microbiome studies across diverse host species. Studies have ranged from assessments of GI microbial richness and diversity to classification of novel microbial lineages. Assessments of the “normal” state of the GI microbiome composition across multiple host species has gained increasing importance for distinguishing healthy versus diseased states. This study aimed to determine baselines and trends over time to establish “typical” patterns of GI microbial richness and diversity, as well as inter-individual variation, in three populations of western lowland gorillas (*Gorilla gorilla gorilla*) under human care at three zoological institutions in North America. Fecal samples were collected from 19 western lowland gorillas every two weeks for seven months (*n* = 248). Host identity and host institution significantly affected GI microbiome community composition (*p* < 0.05), although host identity had the most consistent and significant effect on richness (*p* = 0.03) and Shannon diversity (*p* = 0.004) across institutions. Significant changes in microbial abundance over time were observed only at Denver Zoo (*p* < 0.05). Our results suggest that individuality contributes to most of the observed GI microbiome variation in the study populations. Our results also showed no significant changes in any individual’s microbial richness or Shannon diversity during the 7-month study period. While some microbial taxa (*Prevotella*, Prevotellaceae and *Ruminococcaceae*) were detected in all gorillas at varying levels, determining individual baselines for microbial composition comparisons may be the most useful diagnostic tool for optimizing non-human primate health under human care.

## 1. Introduction

The richness (the number of distinct microbial taxa in a biological sample) and Shannon diversity (which incorporates richness as well as the relative evenness of representation of taxa) of a host’s microbiome are affected by a multitude of factors including the host’s age, diet, health, phylogeny, season, and sex [1,2,3,4,5,6,7,8,9,10]. Host phylogeny and dietary niche are two of the strongest drivers not only of a given host’s gastrointestinal (GI) microbiome but also the evolutionary development of the microbiome associated with a given species [3,4,10,11,12,13]. Previous research has focused on differences among various host species, highlighting inter-species differences associated with ecological environment or evolutionary niche [2,8,10,11,12,13]. Other studies have assessed disease or life stage effects, such as development of the microbiome within human infants and non-human primates [7,9,14,15,16,17,18]. These studies, however, were frequently restricted by which host species were included, a low number of individuals sampled over time, and/or short sampling periods.

While microbial differences (i.e., beta diversity, the measure of how dissimilar two microbial communities are to each other) among host species are increasingly characterized, inter-individual differences among conspecific hosts are still poorly understood. In humans, the GI microbiome is highly variable among individuals [19,20,21]. However, few other host species have large enough datasets that include longitudinal sampling from sufficient individuals to determine whether their microbiomes display the same plasticity across individuals as the human microbiome [1]. This lack of longitudinal datasets for non-human species also makes it difficult to fully understand the “typical” state for a given host species. Sufficient data to characterize typical temporal variation over time for an individual within a given species are necessary to effectively apply microbiome data to assess the health of representative healthy, weaned individuals exhibiting no clinical conditions. The gut microbiome is inextricably tied to most/all host life processes, but the causal versus correlative nature of the relationship is dynamic and, indeed, still unknown for many conditions. As such, while the gut microbiome may provide useful indicators of several life stages and health conditions, its current utility is currently limited.

To address this gap in non-human primates, we compiled and analyzed a longitudinal fecal microbiome dataset for western lowland gorillas (*Gorilla gorilla gorilla*) under managed care. To the authors’ knowledge, this dataset represents the largest number of individual gorillas sampled over the longest time duration to date. Based on previous work in humans and rats [19,20,21], we hypothesized that host identity and home institution (including its specific management and diet program) would affect the richness and beta-diversity of the western lowland gorilla microbiome. Specifically, we predicted that microbiome characteristics between individuals at the same institution would be more similar than between individuals from different institutions. We also expected to detect changes over time in both the richness and beta-diversity of individual microbiomes due to seasonal changes in their respective environments. While individuals did not experience any form of diet change throughout the study period, home institutions were located in temperate regions and therefore underwent seasonal climate shifts (with Denver Zoo experiencing the most extreme seasonal changes of the three institutions). Temperate regions have been documented to have seasonal turnover of soil microbial populations, specifically in Colorado forests [22]. Bornbusch et al. [23] also found that lemur GI microbiomes displayed seasonal fluctuations under human care. Seasonal changes in wild gorilla GI microbiome composition were previously documented by Gomez et al. [10], although these changes were thought to be primarily driven by dietary changes rather than environmental changes.

## 2. Materials and Methods

This study included 19 western lowland gorillas (*Gorilla gorilla gorilla*) housed at three different zoological institutions in North America: Denver Zoo (DZ) in Denver, Colorado (three males, three females); Riverbanks Zoo (RB) in Columbia, South Carolina (two males, three females); and North Carolina Zoo (NC) in Asheboro, North Carolina (five males, three females). Individual ages ranged from 2 to 45 years. Two females were humanely euthanized during the collection period (DZF3 and NCF1), due to endometrial cancer and complications associated with a sudden perforated bowel due to a suspected foreign body, respectively. The first female was under long-term care for poor health and survived only three sampling rounds before passing; the second female was healthy and contributed to seven rounds of collection before her emergency bowel perforation. A third female, RBF1, was pregnant and gave birth during the study period; her infant was not included in the study sampling. All individuals were maintained on their home institution’s standard diet and no overall diet changes were implemented during or prior to the study period (Table 1).

Denver Zoo is an 80-acre facility with a high altitude (5285 ft) highland steppe climate. The NC Zoo is a 500-acre facility and considered the world’s largest natural habitat zoo. This facility is a lower altitude facility (627 ft) located in a temperate deciduous forest biome. Like the NC Zoo, the Riverbanks Zoo is a lower elevation facility (167 ft) in a temperate deciduous forest biome. All three institutions include both indoor and outdoor exhibit spaces, and all individuals had access to both indoor and outdoor spaces throughout the study. The outdoor exhibit spaces consisted of “natural” environments (i.e., soil and vegetation), while indoor spaces consisted of cleanable surfaces (i.e., concrete) and bedding substrates (i.e., woody mulch, wood wool, straw).

Fecal samples were collected from individual gorillas every two weeks from 1st February 2018 until 31st August 2018 (7 months; approximately 15 samples per individual for a total *n* = 256). Delays in initial sampling as well as health issues or behavioral issues (i.e., unwilling to shift to collect samples) during the collection periods led to some individuals contributing fewer than 15 samples across the study period. Samples were collected from the floor of the indoor enclosure within 30 min of defecation and placed in Whirl-Pak ^®^ bags (Nasco, Fort Atkinson, WI, USA) using sterile tongue depressors and nitrile gloves (SensiCare^®^Silk powder free nitrile gloves, Medline Industries^©^, Northfield, IL, USA) to minimize contamination from the environment or collector. The samples were immediately transferred to a −80 °C freezer at each zoo, where they were stored until transfer on dry ice to NC State University (Raleigh, NC, USA) for DNA extraction.

### 2.1. DNA Extraction

DNA extractions were performed in two batches at North Carolina State University. The first batch included all fecal samples collected between February and May 2018, and the second batch included samples collected between June and August 2018). Immediately before extraction, the outside layer of each fecal sample was shaved away with a sterile razor blade to expose the interior of the fecal sample, which had no contact with the environment during collection. A 0.25 g sample was then removed from the inner portion of the feces for DNA extraction. DNA was extracted using the QIAGEN DNEasy PowerSoil kit from QIAGEN (Hilden, Germany) with modifications to the manufacturer specifications, as previously described [7]. DNA elutions were quantified with a Nanodrop 2000 (Raleigh, NC, USA) and stored at −80 °C until sequencing.

### 2.2. Amplicon Sequencing

Standardized DNA aliquots were shipped overnight on dry ice to the Primate Microbiome Project (Nebraska Food for Health Center, Lincoln, NE, USA). All samples were prepared for 2 × 250 pair-end sequencing on the Illumina© MiSeq^TM^ (San Diego, CA, USA). The 16S V4 region of rRNA was amplified and sequenced as previously described using the 16Sf and 16Sr gene specific primers [22]. After sequencing, the raw fastq files were used to produce operational taxonomic unit (OTU) tables using Quantitative Insights into Microbial Ecology (QIIME; open source www.qiime.org), using a 97% similarity cut-off and the SILVA132 database to assign taxonomy at the genus level [24,25,26]. Low quality sequences and reads shorter than 50 base pairs were removed from the dataset. All sequence libraries were rarefied to 10,000 reads per sample to remove any low-yield samples that could skew the data [27]. 

The rarefied dataset was analyzed using the vegan package for the open-source statistical software R (version 1.1.463) and the SciPy and Seaborn packages for Python [28,29]. All analyses were conducted on pre-taxonomy OTU counts to avoid any bias that might result at different taxonomic levels. A heat map of relative abundance and longitudinal composition plots were generated using OTU assignment at the L6 (i.e., genus) level of classification. Samples collected from individual DZF3 while they were sick and euthanized were excluded from the statistical analysis. All samples collected from NCF1 were included in the analysis, because this individual became abruptly sick during the study and was healthy up till that point. Individual RBF1, who was pregnant, was also included because pregnancy is considered a desirable condition in a healthy population. Alpha diversity (richness, Shannon diversity, and Faith PD) and beta diversity (Bray–Curtis dissimilarity, Jaccard index) distance metrics were calculated using QIIME2. Kruskal Wallis and ANOVA tests were used to compare alpha diversity metrics across host and environmental variables.

The effects of host identity, season, sex, housing group and institution on microbial community composition, as measured by Bray–Curtis and Jaccard dissimilarity metrics, were determined using ADONIS (single permanova, 999 permutations) [30]. Spearman correlation was used to assess the relationship between individual ages and alpha diversity metrics. All samples were analyzed together to compare the differences among host metadata parameters, and within each separate institution to determine whether individual institutions had effects that were masked in the full dataset. T-tests were used to assess differences in relative abundance among samples for those microbial taxa that were present at >1% abundance across all samples, to identify dominant lineages within individuals. Significance for all statistical comparisons was determined at *p* ≤ 0.05.

## 3. Results

A total of 256 samples were collected (92 from DZ, 92 from NC, and 72 from RB). Of those, a total of 248 sample extractions contained sufficiently high-quality DNA to be sequenced (84 from DZ, 92 from NC, and 72 from RB). The three samples from DZF3 were then removed and 245, therefore, remained. After rarefaction a total of 196 samples remained for downstream analysis (47 from DZ, 80 from NC, and 69 from RB; see Figure A1 for alpha rarefaction curve of observed OTUs per sample). Details on the number of samples per individual and per institution are provided in Table 2. Only samples from the current study were included in this paper; no external sequences from previous publications were included. This was done due to the rich nature of the current dataset, and to avoid confounding variables associated with previous work including differences in sequencing approach, region of the 16S gene sequenced, and availability of comparable metadata. By restricting analysis to the current longitudinal dataset, we were able to detect both individual clouds as well as potential seasonality of western lowland gorilla microbiomes under managed care.

We identified 10,446 distinct OTUs across all samples, with 9722 of these OTUs classified beyond the domain level (Figure 1 and Figure 2). The mean number of observed OTUs differed significantly among the three institutions (ANOVA, *p* = 0.006), with Denver Zoo displaying the lowest mean number of observed OTUs (DZ = 245.32, NC = 275.03, and RB = 260.91) (Figure 1). However, Faith PD and Shannon diversity were not significantly different when samples were grouped by institution (Faith PD ANOVA *p* = 1.1, Shannon Diversity ANOVA *p* = 0.6). When samples were analyzed based on the individual, Faith PD and Shannon Diversity were not significantly different (Faith PD ANOVA *p* = 9.3, Shannon Diversity ANOVA *p* = 2.13); but individuals differed significantly in the number of observed OTUs (Observed OTUs ANOVA *p* = 0.08).

We detected significant differences in Shannon diversity associated with host identity (ANOVA, *p* = 0.004), which were most likely driven by the dominance of specific bacterial taxa in different individuals. There were no significance differences in Shannon diversity when samples were grouped by institution (ANOVA, *p* = 0.63). All individuals in this study hosted similar levels of *Prevotella*, *Prevotellaceae*, and *Ruminococcaceae* (Figure 1, Figure 3). Two RB individuals hosted the highest percentages of “Other” microbes (35%), which include all minor taxa found at <1% relative abundance (Figure 2). Individual DZM1 (DZ Male 1) hosted significantly more Treponema 2 (22%) than the other individuals of the study when assessed at absolute maximum of means (Ttest, *p* = 5.447 × 10^−10^). Only Denver Zoo individuals exhibited significant differences in microbial richness and diversity across the length of the study (ANOVA; Bray Curtis DZ *p* = 0.003, NC *p* = 0.25, RB *p* = 0.117; Jaccard DZ = 0.002, NC = 0.169, RB = 0.20).

Analysis of gorillas within each institution revealed that host identity correlated strongly with variation in relative abundance as measured by Bray–Curtis, Jaccard, Weighted Unifrac, and Unweighted Unifrac (ADONIS; *p* = 0.001, R^2^ = 0.193). Denver Zoo individuals displayed more similarity and often overlapping ordinations, indicating that individuals’ microbiomes are more variable at NC and RB (Figure 4). Indeed, ordination plots produced per institution show clustering of samples per individual, indicating that host identity presented the strongest effect on community composition as measured by Bray–Curtis (ADONIS; DZ *p* = 0.021, R^2^ = 0.11; NC *p* = 0.006, R^2^ = 0.10; RB *p* = 0.006, R^2^ = 0.10) and Jaccard distances (ADONIS; DZ *p* = 0.001, R^2^ = 0.10; NC *p* = 0.006, R^2^ = 0.10; RB *p* = 0.006, R^2^ = 0.10).

We also analyzed intra-institutional data using sex, housing group, and age. Sex (i.e., male versus female) significantly affected beta diversity (ADONIS; Jaccard *p* = 0.001, R^2^ = 0.013; Bray–Curtis *p* = 0.001, R^2^ = 0.019; Weighted Unifrac *p* = 0.019, R^2^ = 0.014; Unweighted Unifrac *p* = 0.001, R^2^ = 0.014) but not alpha diversity (Kruskal Wallis *p* = 0.06; mean observed OTUs = 264 for females and 262 for males). Housing group, an indication of the potential for social interaction between individuals, was found to significantly affect alpha (Shannon diversity; Kruskal Wallis *p* = 0.02) and beta diversity metrics (ADONIS; Bray– Curtis, *p* = 0.001, R^2^ = 0.05; Jaccard, *p* = 0.001, R^2^ = 0.04; Weighted Unifrac, *p* = 0.01, R^2^ = 0.034; Unweighted Unifrac *p* = 0.001, R^2^ = 0.046), with the lowest number of OTUs observed within the DZ1 housing group (Table 2) (Kruskal Wallis *p* = 0.02; DZ1 = 236, DZ2 = 254, NC1 = 275, RB1 = 263, and RB2 = 253). Age was also found to significantly affect beta diversity metrics (ADONIS; Bray–Curtis *p* = 0.002, R^2^ = 0.012; Jaccard *p* = 0.001, R^2^ = 0.01; Weighted Unifrac *p* = 0.03, R^2^ = 0.011; Unweighted Unifrac *p* = 0.004, R^2^ = 0.011). A weak correlation was found between age and alpha diversity metrics (Spearman correlation; Shannon diversity *p* = 0.004, correlation = −0.20; Faith PD *p* = 0.02, correlation −0.17, Observed OTUs *p* = 0.008, correlation −0.19).

## 4. Discussion

Here we present the largest and longest consecutive fecal microbiome collection for western lowland gorillas under managed care to date. Host identity significantly affects both the alpha and beta diversity of GI microbiomes across institutions. While microbial taxonomic abundance was similar among individuals within the same institution, no two individuals’ microbial communities were statistically similar. Our results suggest that gorillas display a strong individual identity with regard to GI microbial membership. This finding is consistent with previous research demonstrating that humans have high inter-individual variation, with each individual hosting their own “cloud” of variance [19,20,21,31]. It should be noted that, while individual gorilla microbiomes appear to exhibit some temporal dynamicity (Figure 4), no significant change in intra-individual richness or beta diversity over time was detected except within the Denver Zoo population. Our results suggest that individual gorillas maintained stable GI microbiomes over the 7-month study period.

While individual GI microbiomes maintain stability, we detected significant differences in the relative abundance of several microbial taxa among individual gorillas. For example, individuals DZM1 and DZF3 hosted the highest abundances of *Treponema 2* (22% and 23%, respectively, compared to a range of 5.2%–16% for all other individuals). The *Treponema* genus is a diverse group of microbes that can cause diseases such as syphilis, yaws, and pinta in humans [32]. Not all treponemes have been linked to diseases within humans: in fact, some species are suggested to be normal flora of the human microbiome [32]. Treponemes have also been identified in wild populations of western lowland gorillas [10]. Therefore, its presence within the western lowland gorilla microbiome does not necessarily indicate a potential disease state. However, these findings are interesting since DZF3′s illness required her to be removed from the statistical portion of this study.

Social contact has been linked to similarities in microbiome richness and diversity in baboon populations, sometimes superseding genetic relatedness as a predictor for microbial membership of an individual’s microbiome [33,34]. DZM1 was the alpha male of the family troop that also housed DZF3, and the resulting association may explain why both individuals’ levels of Treponema 2 were high. Indeed, Tung et al. [33] showed that social partners share more similar microbial lineages, at more similar abundances, compared to individuals not in social contact. Conversely, individual RBM1 (a singly housed male) hosted the highest relative abundance of *Anaerovibrio* (2.9%) and *Lactobacillus* (3.8%) compared to other RB individuals. This individual also displayed the lowest abundances for *Prevotella* 1 (0.9%) and *Prevotella* 9 (2.7%) out of all individuals assessed in this study. These discrepancies in RBM1 may reflect a lack of direct social contact, which may have allowed other (environmental) factors to exert a stronger/stochastic influence on his GI community composition. All other individuals sustained social contact throughout the study, allowing for the transfer of microbial content between individuals.

Gorillas at all three zoos hosted similar major bacterial taxa, although relative abundance varied across individuals. This finding is similar in nature to the findings of Campbell et al. [11], who found host lifestyle (i.e., under human management compared to wild) to be a strong predictor of GI microbial composition. However, the current study did not include individuals with varying lifestyles; therefore, individual identity and geographic location became the most prominent drivers of GI microbiome differences in our study populations. Several of the most prominent taxa detected in this study have been previously detected in gorilla GI microbiomes. For example, *Prevotellaceae* and *Ruminococcaceae* are associated with fiber degradation, which is very important to gorillas and other folivores [14,35,36].

Individuals at DZ hosted a greater relative abundance of *Prevotella*, with the youngest individual DZF1 having the highest relative abundance of all individuals studied. The prevalence of *Prevotella* (a carbohydrate generalist) may reflect differences in dietary management between DZ and the other two institutions (Table 1). All three institutions feed similar diet items, with the majority of individual diets comprising leafy greens (romaine, green leaf, kale, etc.) and vegetables (peppers, carrots, broccoli, etc.). However, one noteworthy dietary difference among the three institutions was the inclusion of a commercial complete feed and alfalfa hay at DZ. Additionally, DZ was the only institution that did not feed fruit as a consistent portion of the daily diet.

*Prevotella* is a dominant genus in the human microbiome. The genus has been detected at lower abundances in wild and semi captive non-human primate populations and is found in greater abundance in non-human primates under human care where diets more closely resemble modern “western” human diets compared to wild type diets [2,11]. Changes in *Prevotella* abundance have been previously associated with a change in the quantity and types of dietary polysaccharides consumed [2]. However, the utility of *Prevotella* for assessing host health has not yet been studied. Since *Prevotella* is present in wild populations, the presence alone of *Prevotella* is unlikely to predispose managed populations to diseases or to indicate either the managed or disease status of an individual [12,35,36,37,38,39]. However, it is possible that the relative abundance of *Prevotella* may correlate with different health or disease states. Therefore, further characterization of the relative abundance and species of *Prevotella* present across a larger longitudinal dataset, including managed and wild populations across varying habitats, is needed to better understand the effects of *Prevotella* in managed populations of gorillas. Because prior non-human primate research includes small sample sizes, prior *Prevotella* data may only reflect individual cloud differences.

In addition to the individual, environment and diet also can influence GI microbial richness and beta diversity [10]. Each zoo varied in habitat design as well as daily schedule (e.g., length of time individuals spent outside) and diet. The environments also varied based on regional differences. For example, RB and NC are located in similar climate zones (i.e., relatively high temperature and humidity, with low seasonal variation), while DZ is located in a steppe climate and experiences more drastic seasonal changes in weather and temperature. Due to weather conditions, DZ gorillas experienced limited access to their outdoor habitat and more seasonal changes in the availability of certain diet items throughout the study period. While the DZ populations still had some access to outside spaces, the habitat conditions shifted dramatically in response to seasonal weather patterns. In contrast, the gorillas at NC and RB had nearly year-round access to their outdoor habitat space and experienced less variation in diet. Seasonal changes in environment and diet may therefore explain why significant temporal changes in microbial diversity in DZ individuals were detected. While the diet did not change seasonally per se at any of the three zoos, the sources and types of fruit and vegetables available via food vendors did change and therefore some seasonal variations were difficult to quantify. Additionally, the browse species and quantities available also differ seasonally at each institution. Previous work indicated that four wild gorilla populations in four different regions within the Central Africa Republic exhibited variations in microbiome species richness resulting from exposure to different environmental bacterial species [35] as well as potentially different dietary plant species. However, this research was reported on the level of group and not individual, as samples from each region were analyzed together [35]. Gomez et al., 2015 [35] also found group interaction to be a predictor of differences in GI microbiome richness and diversity, a result supported by the findings of this study. Additionally, by analyzing the dynamics of the microbiome within a given individual, our research has added further insight to how beta diversity metrics can be affected not only by the location of the individual but also by diet, sex, age, and social interaction. Further research studying managed populations across more diverse environmental and dietary conditions will help to further elucidate the potential correlations among diet, environment, and microbial community composition.

It is important to note the large number of DZ samples (*n* = 50) deleted due to low levels of high-quality DNA and rarefaction compared to both NC (*n* = 2) and RB (*n* = 1). The majority of these DZ samples (*n* = 42) were removed from our data during rarefaction. While rarefying data is not supported by all microbiome analysist and statisticians since it eliminates potentially important data [40,41], we included the step to remove any low-yield samples that might bias the dataset [27]. Samples often do not pass rarefaction because the DNA was degraded (e.g., due to environmental exposure or gastrointestinal illness) or because there was not enough microbial content (particularly in geriatric or sick animals because they have not been eating). However, illness does not explain the high number of DZ deletions. This phenomenon more likely reflects the extreme seasonality in DZ (i.e., DNA degraded under the unique environmental conditions there compared to NC and RB) or dietary differences. Specifically, in contrast to NC and RB feeding regimes, gorillas at Denver Zoo received weighed daily quantities of browse and alfalfa—both of which may contain secondary plant compounds (e.g., tannins, saponins) that can interfere with DNA amplification [42,43].

## 5. Conclusions

To date, this dataset presents the largest and longest consecutive sample collection for microbiome study in western lowland gorillas (*Gorilla gorilla gorilla*) under managed care. By conducting a longitudinal study, the authors were able to not only compare across but also within individuals to establish an understanding of the healthy baseline dynamics within and across populations. Using standard microbiome analysis, we determined that host identity and institution significantly affect both the richness and beta diversity of the gorilla microbiome. These effects may be explained by social interactions and (dis)similarities in environment and diet across institutions. In contrast, significant temporal variations in GI community membership and structure were found to be strongly associated with seasonality and dietary changes.

Our findings demonstrate that individual gorillas under managed care host distinct GI microbiome clouds: no two individuals are the same, although individuals may exhibit institutional similarities. These results also highlight the importance of longitudinal sampling, as time-associated variation cannot be detected in single time point samples. More extensive longitudinal datasets are needed to assess host baselines and healthy “normals” in order to leverage GI microbiomes for non-invasive diagnostics. Similarly, the classification of currently unranked taxa may facilitate the correlation of microbial membership to host age and health/disease status.

## Figures and Tables

**Figure 1 animals-11-03399-f001:**
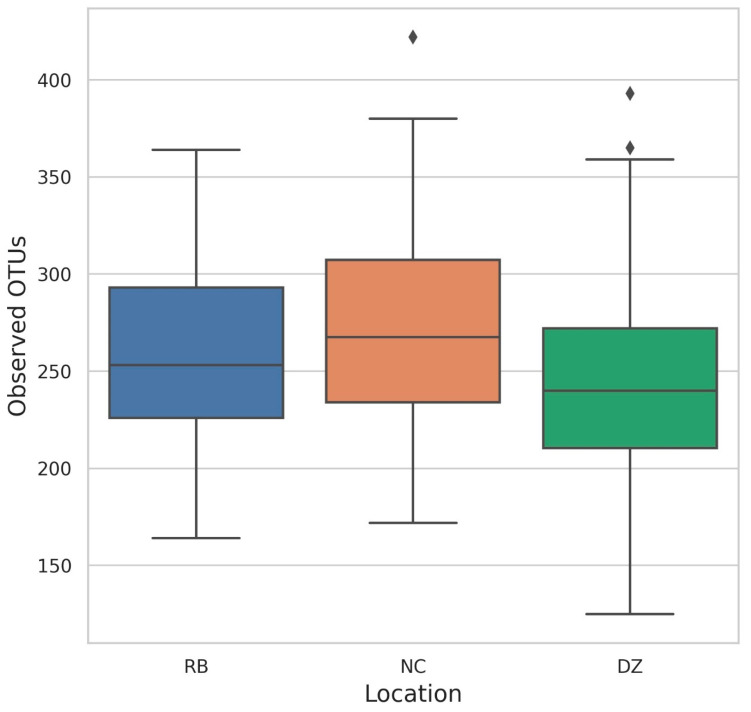
Reported number of observed OTUs (i.e., number of distinct microbial operational taxonomic units identified at the genus level, per sample) in western lowland gorilla (*Gorilla gorilla gorilla*) fecal samples collected from Denver Zoo (DZ), North Carolina Zoo (NC), and Riverbanks Zoo (RB). Outliers indicated by ♦.

**Figure 2 animals-11-03399-f002:**
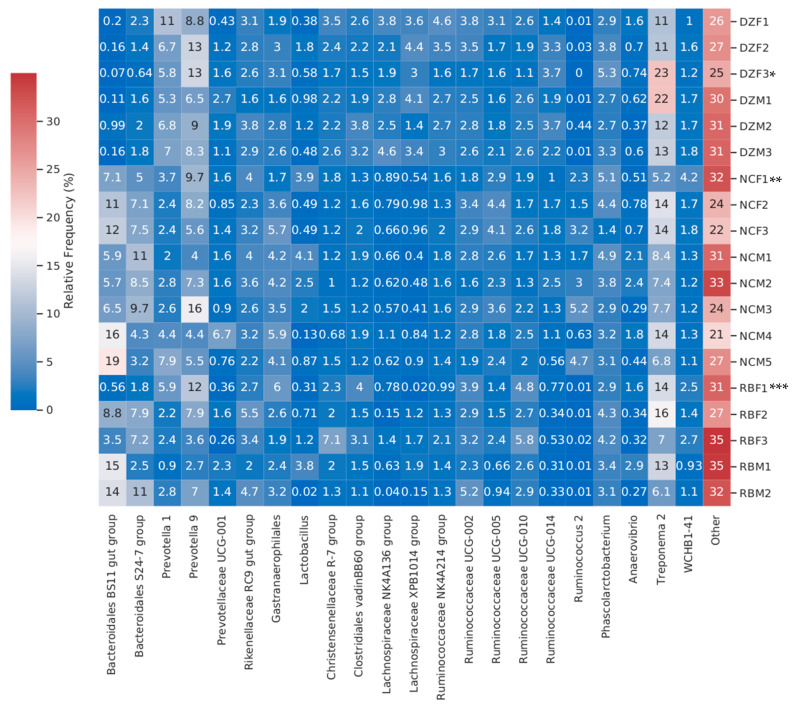
Heat map (with color legend) of major bacterial taxa (i.e., those occurring at >1% relative abundance) in western lowland gorilla (*Gorilla gorilla gorilla*) fecal samples collected from Denver Zoo, North Carolina Zoo, and Riverbanks Zoo, averaged across time points per individual. * DZF3 female euthanized due to long term sickness excluded from statistical analysis, ** NCF1 female was euthanized after emergency perforated bowel but included in analysis due to prior health, *** RBF1 female was pregnant during study. DZ = Denver Zoo (Denver, CO); NC = North Carolina Zoo (Asheboro, NC); RB = Riverbanks Zoo and Garden (Columbia, SC); M = Male and F = Female; 1–5 = individual number provided to animal at each institution; Other category on the y axis represents OTUs found at lower than 1% in relative abundance in an individual’s GI microbiome.

**Figure 3 animals-11-03399-f003:**
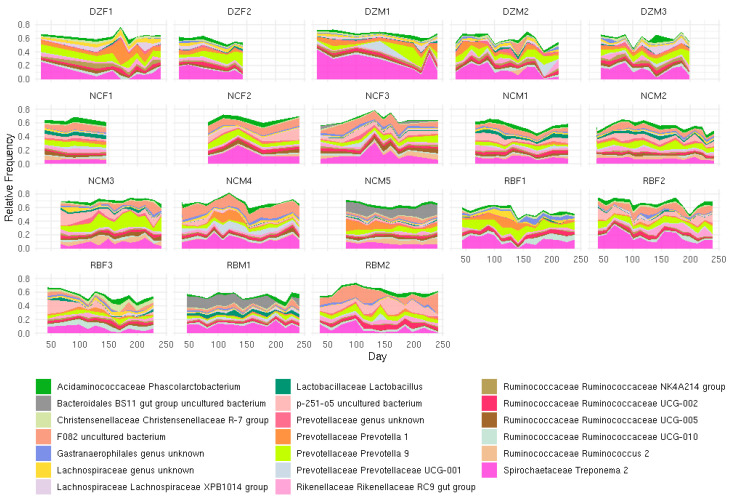
Longitudinal variation in relative abundance of top 20 microbial taxa detected in each individual western lowland gorillas (*Gorilla gorilla gorilla*) through entirety of collection period at Denver Zoo (DZ), North Carolina Zoo (NC), and Riverbanks Zoo (RB), M = Male and F = Female; 1–5 = individual number provided to animal at each institution. Only Denver Zoo samples showed significant change over time ANOVA Bray Curtis DZ *p* = 0.003, NC *p* = 0.25, RB *p* = 0.117; ANOVA Jaccard DZ = 0.002, NC = 0.169, RB = 0.20.

**Figure 4 animals-11-03399-f004:**
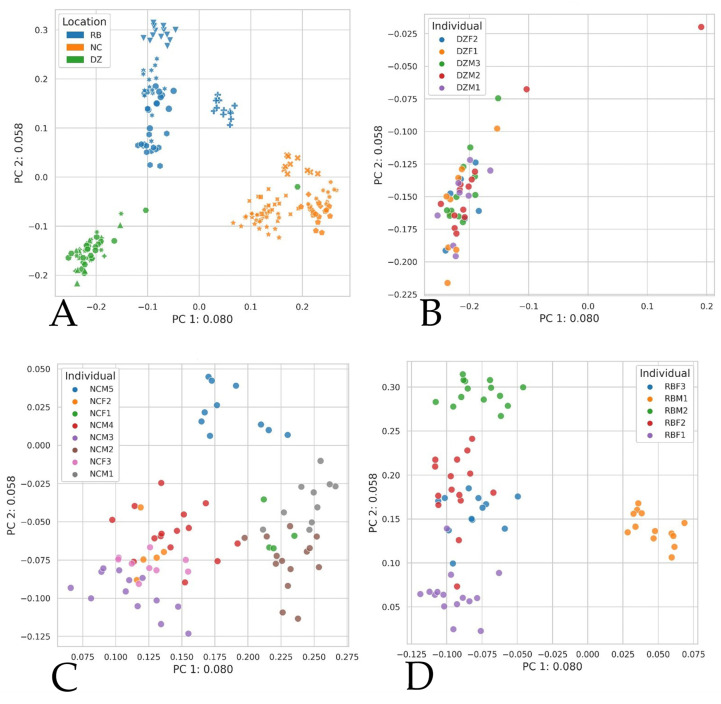
Principal coordinates analysis based on Jaccard index pairwise distances among western lowland gorilla (*Gorilla gorilla gorilla*) fecal samples collected at Denver Zoo (DZ), North Carolina Zoo (NC), and Riverbanks Zoo (RB). (**A**) All institutions together, (**B**) DZ individuals, (**C**) NC individuals, (**D**) RB individuals. Individual animals have sex indicated in their identifier via F = female and M = male. Housing group data not presented although for reference, NC = one group; DZ Group 1 = DZM1, DZF1 & DZF2, DZ Group 2 = DZM2 & DZM3; RB Group 1 = RBM1, RB Group 2 = RBM2, RBF1-RBF3. ADONIS Jaccard DZ *p* = 0.001, R^2^ = 0.10; NC *p* = 0.006, R^2^ = 0.10; RB *p* = 0.006, R^2^ = 0.10.

**Table 1 animals-11-03399-t001:** Breakdown of western lowland gorilla (*Gorilla gorilla gorilla*) diets, as fed by each zoological institution, related as percent (%) of feed item included in institutional diets.

Institution ^1^	Leafy Green Vegetables	Celery ^2^	Other Vegetables	Fruit	Primate Biscuit ^3^	Browse	Alfalfa
DZ	40	11.7	20	na	1.8	15	11.5
NC	66	na	27	7.0	na	variable	na
RB	60	na	33	7.0	na	variable	na

**^1^** DZ, Denver Zoo (Denver, CO); NC, North Carolina Zoo (Asheboro, NC); RB, Riverbanks Zoo and Garden (Columbia, SC); ^2^ Celery was listed separately due to the large quantities offered daily. Celery was also offered at NC and RB but not every day; ^3^ Mazuri Low Starch Primate Biscuit (Mazuri Exotic Animal Nutrition, PO Box 66812, St. Louis, MO 63166); na: denotes item is not included in the daily ration for individuals at that institution. Bold: zoo abbreviation.

**Table 2 animals-11-03399-t002:** Individual list for the nineteen western lowland gorillas (*Gorilla gorilla gorilla*) from Denver Zoo, North Carolina Zoo, and Riverbanks Zoo participating in the fecal microbiome study with total samples collected, total samples remaining after rarefaction and housing group.

Animal ID ^1–3^	Total Samples Collected	Number of Samples after Rarefaction	Housing Group ^4^
DZM1	16	8	DZ1
DZM2	17	13	DZ2
DZM3	16	11	DZ2
DZF1	16	9	DZ1
DZF2	16	6	DZ1
DZF3 *	3	0	DZ1
NCM1	12	10	NC1
NCM2	15	14	NC1
NCM3	13	13	NC1
NCM4	15	15	NC1
NCM5	11	10	NC1
NCF1	7	4	NC1
NCF2	8	5	NC1
NCF3	11	9	NC1
RBM1	14	13	RB1
RBM2	15	14	RB2
RBF1	15	15	RB2
RBF2	15	15	RB2
RBF3	13	12	RB2

**^1^** DZ, Denver Zoo (Denver, CO, USA); NC, North Carolina Zoo (Asheboro, NC, USA); RB, Riverbanks Zoo and Garden (Columbia, SC, USA); **^2^** M = Male and F = Female; **^3^** 1–5 = individual number provided to animal at each institution; **^4^** DZ and RB had their animals housed in two separate housing groups. * Individual excluded from analysis.

## Data Availability

The data that support the findings of this study are available from the corresponding author upon reasonable request.

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
