# Peer review of "Host Identity and Geographic Location Significantly Affect Gastrointestinal Microbial Richness and Diversity in Western Lowland Gorillas (Gorilla gorilla gorilla) under Human Care"

_animals, 2021, doi:10.3390/ani11123399_

Round 1
Reviewer 1 Report
Eschweiler and colleagues propose a study on the variability of the GI microbiome in Gorillas over time and across three zoo institutions. The main objective is to understand better the spectrum of the variability of the GI microbiome according to individuals and zoos to then contribute to implement the GI microbiome as a tool for the health monitoring of zoo gorillas.
If the subject and the dataset are interesting, I have identified five main points:
- You should better explain what is a “normal” or “healthy” GI microbiome. Linked to that it is not clear if the individuals with chronic disease and the pregnant female were included or not. It seems that they were, in this case, it is a problem because they are not “healthy”.
- Given your objective, I really suggest that you add in the method and the result sections a paragraph on the characterization of each zoo institution. The table of the diet and the environment parameters (quickly detailed in the discussion) should appear in the results.
- I do not understand your hypothesis concerning the longitudinal analyses. If it has been shown in free ranging gorillas (you should cite this study, cf detailed comments), it was due to the seasonality of the diet. In the zoo, there is no change in the diet as you mentioned, and probably nor in the environment. It is really important to clarify this point.
- If you want to compare gorillas GI microbiome across zoo institutions, you may want to use published sequence on gorillas GI microbiome to increase your dataset. If not, you could explain why you do not use these dataset.
- It seems that the ASV approach would be better than the OTU approach because it is more accurate.
Moreover, I suggest several complementary references which should be probably cited:
Campbell, T. P., Sun, X., Patel, V. H., Sanz, C., Morgan, D., & Dantas, G. (2020). The microbiome and resistome of chimpanzees, gorillas, and humans across host lifestyle and geography. The ISME journal, 14(6), 1584–1599. https://doi.org/10.1038/s41396-020-0634-2
Gomez, A., Rothman, J. M., Petrzelkova, K., Yeoman, C. J., Vlckova, K., Umaña, J. D., et al. (2016). Temporal variation selects for diet–microbe co-metabolic traits in the gut of Gorilla spp. The ISME Journal, 10(2), 514–526. https://doi.org/10.1038/ismej.2015.146
Narat, V., Amato, K. R., Ranger, N., Salmona, M., Mercier-Delarue, S., Rupp, S., et al. (2020). A multi-disciplinary comparison of great ape gut microbiota in a central African forest and European zoo. Scientific Reports, 10(1), 19107. https://doi.org/10.1038/s41598-020-75847-3
Detailed comments
Introduction
In the first paragraph, it could be interesting to link the state of the art with the temporal variations of the GI microbiome. You claim that previous studies were on a short sampling periods. But, the time was not indicated as a factor of GI microbiome variation at the beginning of this paragraph (first sentence). In free-ranging gorillas, effects of the season has been show on the GI microbiome (Gomez et al. 2016 ISME J).
What is a “normal” GI microbiome? If the GI microbiome influence metabolism, immunity and neurologic function, it could be difficult to be sure that the state observed is normal. I suggest to clarify this point, explaining the limits of the GI microbiome for health monitoring of Gorillas in this context.
Why do you expect changes over time (due to sesonal changes) in a zoo? If the individuals are fed equally over the year, the diet should not be expected to influence GI microbiome over time. If it is due to changes in temperature/hygrometry, potentially affecting the health, it should be clarified. I suggest to really clarify the interest to have the time as a factor on GI microbiome in this context.
In the introduction, you should probably also cited Campbell et al. (2020) who analyzed also variations of the GI microbiome between free ranging and zoo gorillas and chimpanzees, and Narat et al. (2020) who proposed to analyze better the context of “wild” and “captive” conditions when studies look at variability of the GI microbiome across different environment. I suggest you analyze the environment/diet/infrastructures available for the gorillas. Without that point, you propose that each zoo is “just” different location/management/diet with no precision on theses differences, except in the discussion, but I suggest to consider it as a result.
Material and methods
You have samples from two individuals with illness and one pregnant female. So, it is not clear to see the link with your objective: have a better understanding of the “normal” GI microbiome. You should clarify this point both in the introduction and the material and methods. Morevoer, if these individuals were under veterinary treatment, you should explain these treatments because it could also affect GI microbiome.
The table 1 propose a short characterization of each institution only based on the diet. However, we do not know if individuals are only indoors, have an outdoor access with soils, natural vegetation, water… ? See Narat et al. 2020 for an example of how characterized zoo institutions. Moreover, the characterization of this environment, diet etc could be considered as a result and not as a method.
10,000 reads seems a bit low… could you please show rarefaction curves?
Moreover, ASV is a finest way to analyze that compared to OTU. Please consider to use ASV and not OUT in your analyses.
Because several parameters are available and complementary to analyze alpha and beta diversity, you could add OTU/ASV richness and Faith PD index for the alpha diversity and weighted and unweighted unifrac distance for the beta diversity.
Results
Please show in the maint text or in supplementary information the rarefaction curves
In the method you indicate 208 samples after rarefaction and in the results 203… please clarify.
Moreover, you indicate the same kind of information in the method and the result. You could choose a section and indicate these information only once.
1399 microbial genera means identification at the genus level from OTU? Please Clarify
Moreover, it could be interesting to have the proportion of OTU (or ASV if you follow my suggestion) which have been identified at the genus level.
Figure 4: please indicate the percentage on the PC1 and PC2 axis, without that we have no idea about the amount of variability that these two axes indicate. Moreover it seems that the analyses shown in Figure 4 were not described in the method section
When you indicate a result for a statistical test, you could write what was the test, even if it is indicated in the method section.
Given your objective: comparing gorilla GI microbiome across several zoo institutions, why have you not used the published sequences available to increase your dataset? If it does not work for the question of longitudinal analyses (for which you need a better explanation about why this approach is needed), your result show also a “simple” comparison between zoo institution. Thus, it could be interesting to use published sequences…or to explain why you do not use it.
Discussion
Is this surprising that gorillas maintain a stable GI microbiome along 7 months given that the diet does not change? This point in your study is really not clear
All the paragraph on zoo environment should be in result, and you could add a method section and explain better what were the parameters you considered to characterize the zoo environement.
What about the individuals with chronic disease and the pregnant female? You consider them in the general analyses or not? It yes, it is strange because you said that youy wanted to have the “normal” or “healthy” GI microbiome. If no, clarify this point.
Finally, I really encourage the authors to consider these suggestions to improve their manuscript, which could be a very interesting contribution to
Reviewer 2 Report
Review: Animals-1407941 “Host Identity and Geographic Location Significantly Affect Microbial Richness and Diversity in Western Lowland Gorillas (Gorilla gorilla gorilla) Under Human Care”
This article analyzes and compares fecal flora of captive western lowland gorillas in three different zoological institutions collected biweekly over a seven month period. Microbiome investigations have many potential applications, including contribution to knowledge on biology and ecology of species and assessment of health at individual and population levels. This manuscript describes research that is important for building upon foundational microbiome data within the western lowland gorilla species.
Comments and Notes
Line numbers would have been beneficial for this review
All figures need captions revised so that they can stand on their own
There seem to be contradicting statements throughout the paper regarding significant changes in microbial abundance over time. Please clarify and correct.
Abstract
The statement “Significant changes in microbial abundance over time were observed only at Denver Zoo” does not appear to be supported in the results. Please include (or clarify) this information in the results section.
Key Words: Use gastrointestinal instead of or in addition to GI here. Maybe add feces?
Materials and Methods
First paragraph: Regarding the two individuals that died and the pregnant individual-It would be useful to identify these animals up front and within the figures as they represent additional variables and could potentially have some differences in their microbiomes compared to other individuals with whom they share housing. This may have not been the case
Was there any cause identified in the animal with the bowel perforation? This information would rule out for the reader, any presence of underlying illness
Second paragraph: replace “less than” with “fewer than”
2.2 Amplicon Sequencing-first paragraph: should the 208 be 203?
Figure 1: Needs standalone caption. Y axis does not explain what the value is
Table 2: Needs standalone caption. Animal ID’s are not adequately or consistently explained in caption (i.e. certain letters included in zoo ID’s are not present in the ID’s in the table.).
Results
Did age or sex have any effect? If analyzed, please state. Or discuss if this was not explored and why.
Paragraph 3: Are any of the differences concerning the “Other” category and Treponema 2 statistically significant?
Figure 2: Needs standalone caption. Need to define “Other” in the figure (only done in the text). Need to specify number of time points here and that collections were performed every two weeks.
Figure 3: Needs standalone caption. Need to specify that data are showing variation of abundance over time and that samples were collected every two weeks. Showing the increments of 50 days is somewhat misleading.
Paragraph 4: Unless I missed it-Is this where the data are supposed to show that “Significant changes in microbial abundance over time were observed only at the Denver Zoo”? It’s not clear.
Figure 4: Need standalone caption. It might be helpful in this figure to mark individuals as being housed together or separately in B-D
Discussion
First paragraph: Again, the statement in the abstract-“Significant changes in microbial abundance over time were observed only at Denver Zoo” in contradicted here. Conversely, the statement “no significant change in intra-individual richness or beta diversity over time was detected.” should be stated in results.
Second paragraph: The argument here is not well-supported in the results section. Were the abundances of Treponema 2 statistically significantly higher in DZM1 and DZF3? Is this even possible to compare with there being only one usable sample in DZF3? And other individuals in the same enclosure and in other zoos also seemed to have somewhat high levels of Treponema.
Was speciation of the Treponema possible?
Paragraph 7: Again a statement about significant temporal changes in microbial diversity in DZ individuals and not in other institutions. Where are these results?
Paragraph 9: Phenomenon.
What evidence do you have regarding illness not being a reason for so many DZ deletions? What aspects of the DZ diet compared to the other zoo diets could have produced inhibitors to such an extent that so many samples had to be removed from analysis? What is your argument for using rarefaction? Maybe subsequent studies should look at different approaches and note differences in outcome? This paragraph in general needs better explanation.
Conclusion
First paragraph: Significant temporal variations?? See above comments
Please adjust conclusions accordingly if changes/clarifications are made.
Author Response
Please see the attached report.

Round 2
Reviewer 2 Report
I am satisfied with the revisions. Thank you!
One question: Maybe I missed it but I did not see criteria for statistical significance in the methods. Please make sure that that is taken care of. For example, you deemed significance for the ANOVA in line 213 at p=0.08 and I don't see anything stated about your chosen p values in the methods.
Author Response
Thank you for the quick second review and finding that missing comment.
We added a line 188 to our text: "Significance for all statistical comparisons was determined at P ≤ 0.05."
Best, always,
Kimberly